# Effects of Antiviral Therapy and Glucocorticoid Therapy on Fever Duration in Pediatric Patients with Influenza

**DOI:** 10.3390/medicina57121385

**Published:** 2021-12-20

**Authors:** Ji Yoon Han, Eun Ae Yang, Jung-Woo Rhim, Seung Beom Han

**Affiliations:** 1Department of Pediatrics, College of Medicine, The Catholic University of Korea, Seoul 06591, Korea; han024@catholic.ac.kr (J.Y.H.); anni79@catholic.ac.kr (E.A.Y.); jwrhim@catholic.ac.kr (J.-W.R.); 2Department of Pediatrics, Daejeon St. Mary’s Hospital, The Catholic University of Korea, Daejeon 34943, Korea

**Keywords:** influenza, glucocorticoid, neuraminidase inhibitor, child

## Abstract

*Background and Objectives*: Considering developing resistance against neuraminidase inhibitors (NAIs) and their adverse reactions, restricted use of NAIs and use of alternative drugs should be considered for treating influenza. Although glucocorticoids (GCs) have been used for severe influenza, their effects on non-severe influenza have rarely been evaluated. This study aimed to evaluate the clinical responses to NAI therapy and GC therapy in pediatric patients with non-severe influenza. *Materials and Methods*: A total of 601 pediatric patients (<19 years of age) diagnosed with non-severe influenza were retrospectively recruited to evaluate the effects of NAI therapy and GC therapy. Post-admission fever duration and hospitalization duration were compared among four patient groups divided by the administered treatment: No therapy (*n* = 52), NAI therapy (*n* = 154), GC therapy (*n* = 123), and Both therapies (*n* = 272). *Results*: In a multivariate analysis with adjustment for confounding variables, the post-admission fever duration was not significantly different among the four patient groups. The post-admission fever duration tended to shorten with increasing age, longer pre-admission fever duration, and incidence of influenza A virus infection and lower respiratory tract infection. The type of administered treatment showed no significant effects on the post-admission fever duration in any subgroups according to patient age, pre-admission fever duration, influenza virus subtype, and clinical diagnosis. *Conclusions:* Symptomatic treatment rather than antiviral or GC therapy seems to be sufficient for patients with non-severe influenza, although the effects of NAI therapy and GC therapy according to their administered time and dose should be further evaluated.

## 1. Introduction

Influenza is a common respiratory tract infection, and annual influenza epidemics recur during winter and spring with occasional pandemics [1]. Adamantanes targeting the M2 ion channel of the influenza virus were developed as antiviral agents several decades ago [2]. However, their use is no longer recommended because of the lack of effects on influenza B virus and increased resistance against influenza A virus since the 2000s [2]. Thereafter, neuraminidase inhibitors (NAIs) possessing antiviral effects on both influenza A and B viruses were introduced and are currently regarded as first-line antiviral agents for the influenza viruses [2]. The rate of resistance to NAIs began to increase during the 2007–2008 influenza season, and >90% of influenza A(H1N1) strains circulating in the 2008–2009 influenza season were resistant to oseltamivir [2]. Although the influenza A(H1N1) pdm09 strain, a seasonal influenza strain since the influenza pandemic in 2009, has a low rate of resistance to NAIs [1], resistances to both NAIs and newly developed antiviral agents, such as protease inhibitors and RNA polymerase inhibitors were reported [2]. Therefore, restricted use of NAIs and use of alternative drugs should be considered for treating influenza.

Glucocorticoids (GCs) have been used to treat severe influenza as an adjunctive drug to NAIs [3]. However, most previous studies have reported no significant therapeutic effects [4]. Low-dose GC therapy seemed to be effective in treating acute respiratory distress syndrome (ARDS) accompanied by influenza [5]; however, the detailed effects of GC therapy according to the type, dose, and administration time of GCs and the age and underlying medical conditions of patients have not been determined. Moreover, the effects of GC therapy in patients with non-severe influenza have rarely been evaluated, although almost all patients with influenza experience mild-to-moderate symptoms [1,6].

NAIs suppress cell-to-cell transmission of influenza viruses by preventing the release of new viruses from infected cells [2]. Consequent reduction of viral load and diminished host immune responses promote symptom resolution. In patients infected by influenza virus, tissue damage and systemic and respiratory symptoms are considered to be caused by host immune responses rather than direct viral invasion [7]. Innate immune cells recognize viral RNA and produce pro-inflammatory cytokines for viral clearance in the early phase of influenza viral infection; however, uncontrolled pro-inflammatory responses may raise the severity of influenza [7]. In this regard, GCs, which suppress the production of pro-inflammatory cytokines [8], may be useful for reducing systemic and respiratory symptoms including fever. If GC therapy improves patients’ symptoms without increasing adverse reactions and secondary complications, mild-to-moderate influenza can be treated with low medication costs and the development of resistance to NAIs can be suppressed. In this study, the clinical responses to NAI therapy and GC therapy were evaluated in pediatric patients with mild-to-moderate influenza between the 2014–2015 and 2019–2020 influenza seasons.

## 2. Materials and Methods

### 2.1. Subject and Data Analysis

The medical records of pediatric patients aged <19 years who were hospitalized in the Department of Pediatrics, Daejeon St. Mary’s Hospital (Daejeon, Korea) between the 2014–2015 and 2019–2020 influenza seasons were retrospectively reviewed. Among them, patients with severe influenza, who required oxygen supplementation, mechanical ventilator care, or intensive care unit (ICU) admission and who had accompanying shock or death, were excluded (Figure 1). Patients who had underlying systemic disorders with an increased risk for infection, for example, neuromuscular diseases, chromosomal anomalies, and organ-transplanted states, were excluded. Meanwhile, those with non-severe influenza were included. In this study, the fever duration was determined as the representative parameter of clinical response to the administered treatment because it was difficult to identify and verify symptom severity in children, while body temperature could be objectively verified. Accordingly, afebrile patients were excluded from this study. Patients in whom NAI therapy or GC therapy was initiated before admission were excluded because their exact fever duration after treatment could not be determined. Prolonged fever before treatment could be caused by febrile illnesses other than influenza; therefore, patients who complained of fever lasting for >7 days before admission were also excluded. Furthermore, patients diagnosed with croup were excluded from this study because the therapeutic effects of GCs on croup have been previously defined [9]. Influenza was diagnosed when influenza A or B virus was identified using a commercial rapid influenza detection test (RIDT) kit (Alere BinaxNOW^®^ Influenza A & B Card, Abbott, IL, USA) or a multiplex polymerase chain reaction (mPCR) test kit (Anyplex™ II RV16 Detection kit, Seegene Inc., Seoul, Korea) for respiratory viruses using a nasopharyngeal swab sample. If influenza A and B viruses were concurrently identified, the patient was excluded. Patients with discrepant results of the RIDT and mPCR tests were also excluded.

The demographic data, including age and sex, and clinical data, including the patients’ symptoms and clinical diagnoses on admission and at the end of treatment, were collected. Administration of NAIs and GCs, duration of GC therapy, and maximum dose of administered GCs were investigated. The pre- and post-admission fever durations were evaluated to determine the clinical responses to the administered treatment: fever duration in days before admission and fever duration in hours after admission. The included patients were divided into four groups according to the administered treatment: No therapy group (including those receiving symptomatic treatment without NAI and GC administration), NAI group (including those receiving NAI therapy without GC therapy), GC group (including those receiving GC therapy without NAI therapy), and Both therapies group (including those receiving NAI therapy and GC therapy). Demographic and clinical characteristics as well as fever and hospitalization durations were compared among the four treatment groups. In addition, the independent effect of the administered treatment on the post-admission fever duration was evaluated in the whole study population and in the subgroups categorized according to the factors expected to affect the post-admission fever duration: clinical diagnosis on admission (upper respiratory tract infection (URI) and lower respiratory tract infection (LRI)), subtype of influenza virus (influenza A virus and influenza B virus), fever duration before admission (>2 days and ≤2 days), and patient age (<5 years, 5–9 years, and ≥10 years). This study was approved by the Institutional Review Board of the Daejeon St. Mary’s Hospital, which waived the need for informed consent (approval number: DC21RISI0008).

### 2.2. Definitions

The influenza season was defined as the period from October to May. NAI therapy was defined when a body weight-appropriate dose of oral oseltamivir was administered ≥10 times or that of intravenous peramivir was administered once or more. GC therapy was defined when any types and doses of GCs were administered once or more. The administered GC doses were converted to dose equivalent to those of prednisolone based on the following anti-inflammatory potencies of GCs: hydrocortisone, 1; prednisolone, 4; methyl-prednisolone, 5; and dexamethasone, 20. The post-admission fever duration was defined as the time from arrival at an emergency medical center or inpatient ward to the time when fever (a body temperature of ≥38 °C) was checked finally, followed by a body temperature of <38 °C maintained for more than 48 h. For hospitalized patients, body temperature was measured using an infrared tympanic membrane thermometer on admission, and then regularly thrice a day (at 6:00, 12:00, and 18:00). When a patient or guardian complained of febrile sensation, body temperature was additionally measured. For patients with fever, body temperature was measured hourly until defervescence was identified. Hospital-acquired infection was defined as fever and respiratory symptoms that developed >48 h after admission or ≤48 h after discharge from previous hospitalization; for hospital-acquired infection, the fever duration was calculated from the time of fever development after >48 h of admission.

### 2.3. Statistical Analysis

For comparisons of demographic and clinical characteristics among the four treatment groups, the Kruskal–Wallis test and chi-square test were used for continuous and categorical variables, respectively. For the whole study population, a Cox proportional hazards model was used for the multivariate analysis to determine the independent effects of the administered treatment on the post-admission fever duration while adjusting for confounding variables. Multivariate Cox regression analyses were also performed in the following patient subgroups: patients diagnosed with URI and LRI; those infected by influenza A virus and influenza B virus; those with pre-admission fever lasting for >2 days and ≤2 days; and those aged <5 years, 5–9 years, and ≥10 years. The SPSS version 21 program (IBM Corporation, Amork, NY, USA) was used for the statistical analyses, and statistical significance was defined as a two-tailed *p*-value of <0.05.

## 3. Results

Between the 2014–2015 and 2019–2020 influenza seasons, a total of 826 pediatric patients were admitted with influenza. Among them, 815 (98.7%) patients with non-severe influenza were identified after excluding 11 (1.3%) patients receiving oxygen supplementation. Seven (63.6%) patients of them had underlying disorders: three with genetic disorders, two with asthma, one with a chromosomal anomaly, and one with cerebral palsy and epilepsy. All of them received oxygen therapy. Although four (36.4%) patients (three with underlying disorders and one without an underlying disorder) were admitted to the ICU and received invasive mechanical ventilation, no patient died of influenza. One patient with a chromosomal anomaly experienced ARDS, received invasive mechanical ventilation, and eventually recovered. Two patients with asthma experienced asthma exacerbations, and the remaining four patients with underlying disorders experienced pneumonia. Among four patients without underlying disorders, three patients, including one receiving invasive mechanical ventilation, experienced bronchiolitis, and the other patient experienced pneumonia.

The administered treatment for influenza was evaluated in 778 patients after excluding 21 patients with underlying systemic disorders, nine afebrile patients, and seven patients admitted with fever lasting for >7 days. Further, 68 and 56 patients who received NAI therapy and GC therapy before admission, respectively, were excluded. Among the remaining 654 patients, 29 patients in whom influenza A and B viruses were concurrently identified and three patients in whom discrepant RIDT and mPCR test results were identified were excluded. A total of 21 patients diagnosed with croup were also excluded. Eventually, 601 patients were included in the statistical analyses (Figure 1).

For the finally included patients, the median age was 4 years (interquartile range: 2–7 years), and 324 (53.9%) were men. The influenza virus was identified on an RIDT in 449 (74.7%) patients, an mPCR test in 107 (17.8%) patients, and on both tests in 45 (7.5%) patients. Influenza vaccination history was recorded in 97 (17.3%) of the 562 patients aged ≥6 months; 69 (71.1%) of them received influenza vaccination for the same influenza season.

### 3.1. Clinical Characteristics of the Four Treatment Groups

Among the four treatment groups, patient age and clinical diagnosis were significantly different: patients receiving GC therapy (GC and Both therapies groups) were younger and more likely to be diagnosed with LRI than those not receiving GC therapy (No therapy and NAI groups). Acute otitis media was more common in the No therapy group than in the other groups (*p* = 0.007, Table 1); however, 76.9% of them were diagnosed on admission. The pre-admission fever duration, overall fever duration, and hospitalization duration were not significantly different among the four treatment groups (Table 1).

### 3.2. Factors Associated with the Post-Admission Fever Duration

A multivariate Cox regression analysis was performed to determine the independent effects of administered treatment on the post-admission fever duration. In the multivariate analysis for the whole study population, the post-admission fever duration tended to be shorter with increasing age, longer pre-admission fever duration, higher incidence of influenza A virus infection than of influenza B virus infection, and higher incidence of LRI than of URI (Table 2 and Figure 2).

However, the type of administered treatment did not have a significant independent effect on the fever duration after treatment (Table 2 and Figure 3).

### 3.3. Subgroup Analyses

The subgroups were categorized according to the patients’ clinical diagnoses (URI and LRI), influenza virus subtype (influenza A virus and influenza B virus), pre-admission fever duration (≤2 days and >2 days), and patient age (<5 years, 5–9 years, and ≥10 years). In each subgroup, a multivariate Cox regression analysis was performed to determine the independent effects of administered treatments on post-admission fever duration: the fever durations after admission of the four treatment groups showed no significant differences in any subgroups (Appendix A).

## 4. Discussion

In this study, the post-admission fever duration was determined as the representative parameter of the clinical response to treatment in pediatric patients with non-severe influenza. We found that NAI therapy and GC therapy yielded no significant effect on the fever duration in an actual clinical setting.

NAI therapy shortened the symptom duration among pediatric patients with influenza in a meta-analysis of well-controlled clinical studies; however, the duration was shortened by only 18 h [10]. Especially in children, the time when influenza-consistent symptoms develop and disappear after treatment might not be determined accurately because children may not appropriately complain of their own symptoms, and their parents are responsible for recognizing their symptoms. Therefore, the therapeutic effects of NAI therapy in actual clinical settings could be less than those observed in well-controlled clinical studies. A previous observational study has actually failed to show a significant reduction in symptom duration after NAI therapy in pediatric patients with influenza [11].

GC therapy facilitates symptom relief by 4–6 h in patients with non-severe respiratory tract infections [12], while approximately 10% of patients with URI received GC therapy in actual clinical settings [13,14]. GC therapy showed more significant effects in patients infected by group A streptococci than in those infected by respiratory viruses [12], and its effect on non-severe influenza has rarely been evaluated. Among the patients with non-severe influenza in this study, the hospitalization duration and post-admission fever duration of the patients receiving NAI therapy, GC therapy, or both therapies were comparable to those of the patients receiving only symptomatic treatment. The multivariate analysis with adjustment for confounding variables showed no significant effects of administered therapies on the post-admission fever duration. The post-admission fever duration tended to decrease with increasing pre-admission fever duration, and the overall fever duration was not significantly different among the four treatment groups. In this study, defervescence was observed within 24 and 48 h after admission in 60.7% and 86.2% of the included patients, respectively. Previously reported reduction of the symptom duration by 18 h with NAI therapy and 4–6 h with GC therapy might not be clinically significant in actual clinical settings. Severe complications have occurred in 0.2–0.5% of patients with influenza in previous studies [1,6], and severe influenza developed in 1.3% of the hospitalized pediatric patients in this study. Considering this low occurrence rate of severe influenza and brief reduction of the symptom duration with NAI therapy and GC therapy, universal NAI therapy or GC therapy should not be applied to all patients with influenza. In accordance with the recommendation of the World Health Organization, NAI therapy should be restricted to patients with a risk for severe influenza [1].

GC therapy in patients with severe influenza increased secondary infection and mortality rather than yielded therapeutic effects [15,16,17,18,19,20]. Meanwhile, recent studies where confounding variables were adjusted reported weak evidence for increasing adverse effects and mortality with GC therapy despite insignificant therapeutic effects [4,21]. Moreover, the therapeutic effects of GC therapy in patients with severe influenza vary according to the time interval between GC administration and symptom development, administered GC dose, and patient age [5,15,16,17,22]. Severe neurologic complications of influenza, including influenza-associated encephalopathy, are thought to be caused by cytokine storm [23], and therefore, GC therapy may be helpful for preventing or treating severe neurologic complications as well as severe pulmonary complications in patients with influenza. However, recent studies failed to show significant results on the effects of NAI therapy and GC therapy on the development of influenza-associated neurologic complications [24,25]. Because neurologic complications occurred within a few days of influenza symptoms, NAI therapy and GC therapy were initiated after the development of neurologic complications in most patients [24,25]. Moreover, early steroid pulse therapy initiated within 2 days of admission did not show a significant effect in pediatric patients with influenza-associated encephalopathy [26]. Influenza viral load peaks around the development of fever, and then, viral load decreases and fever disappears within 5 days without specific therapies in patients infected by influenza virus [27]. Considering that classical genomic effects of GCs begin several hours after GC administration, early GC therapy may suppress host immune responses required for viral clearance during early phases of inflammation [8]. In this regard, GC therapy performed several days after inflammation when some extent of viral clearance is achieved may be useful. Considering enhanced innate immune responses and clinical effects in severe influenza by low-dose GC therapy [5,8], low-dose GC therapy can be helpful.

The decrease in the post-admission fever duration with increasing age in this study seemed to be associated with accumulated immunity against influenza over time, acquired via recurrent vaccination. The influenza vaccination rate in Korean children is higher than that in other countries: 56.3% in children and adolescents aged <19 years and 84.2% in children aged ≤5 years [28]. In this study, 69 (71.1%) of 97 patients aged ≥6 months, in whom influenza vaccination history could be determined, received influenza vaccination during the influenza season. Vaccine-acquired immunity might diminish the therapeutic effects of NAI therapy and GC therapy, and repeated vaccination after infancy might be helpful for reducing the symptom duration of influenza. However, the influence of vaccination on the effects of NAI therapy and GC therapy could not be further evaluated because the number of patients with vaccination records was small. Although the patients with LRI showed shorter post-admission fever duration than those with URI in this study, the pre-admission fever duration was longer in the patients with LRI than in those with URI, and the overall fever duration was comparable between the two patient groups. Because LRI in patients with influenza is more likely to be caused by secondary bacterial infection than by direct viral dissemination to the lower respiratory tract [29], antibiotics administered to patients with LRI during hospitalization might reduce the post-admission fever duration. However, 85.5% of the whole study population in this study received antibiotic therapy. Although the patients with LRI received antibiotic therapy more frequently than those with URI (95.9% vs. 83.9%, *p* = 0.002), a multivariate analysis including antibiotic therapy as a confounding factor showed no significant effect of antibiotic therapy on the post-admission fever duration (hazard ratio = 0.854, *p* = 0.191).

This study had some limitations. As mentioned above, the time of symptom development and disappearance might not be accurate in pediatric patients, and the severity and duration of symptoms before admission could not be investigated owing to the retrospective nature of this study. To overcome these problems, we included patients only when NAI therapy and GC therapy were initiated after admission. Considering the short febrile period in patients with influenza, delaying specific treatment by one or two days might have a significant impact on its therapeutic effects. Future studies should be planned to determine the time of symptom development and resolution as well as treatment initiation and completion accurately. The frequencies of some symptoms, including headache and rash, were significantly different among the four treatment groups. However, they were not considered in the multivariate analysis to determine therapeutic effects of NAI and GC because a small number of patients complained of the symptoms, and we considered their influence on the effects of NAI therapy and GC therapy negligible. The effects of the time, dose, and duration of GC therapy could not be evaluated because various strategies of GC therapy were performed according to the attending physician’s decision. NAI therapy and GC therapy were not performed in accordance with a pre-defined guideline, and therefore, unknown confounding factors might exist. Well-controlled clinical studies adopting a reliable and easily applicable therapeutic guideline in actual clinical settings are necessary.

## 5. Conclusions

In conclusion, NAI therapy and GC therapy showed no significant effects on the fever duration in the pediatric patients with non-severe influenza. Appropriate symptomatic treatment rather than specific antiviral and immune modulating therapies seems to be sufficient for patients with non-severe influenza, considering the low occurrence rate of severe influenza and the brief effects of NAI therapy and GC therapy.

## Figures and Tables

**Figure 1 medicina-57-01385-f001:**
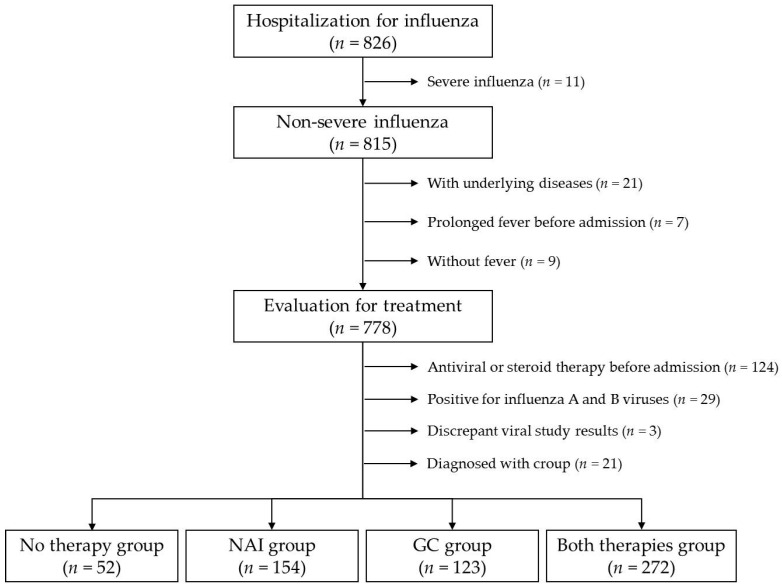
Flowchart of study population selection.

**Figure 2 medicina-57-01385-f002:**
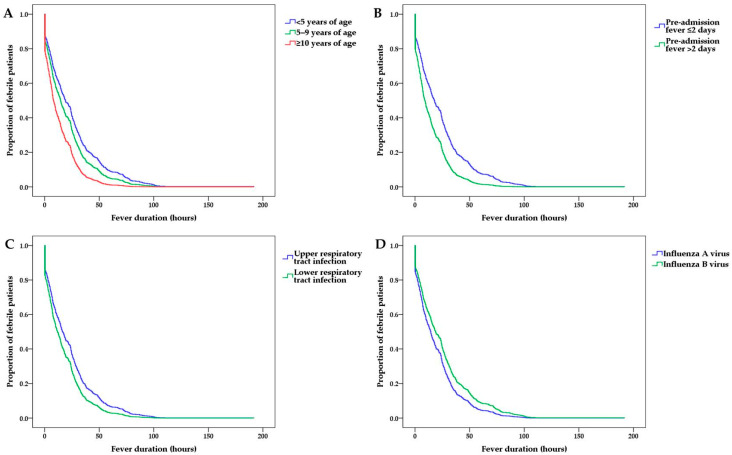
Post-admission fever duration according to (**A**) patient age, (**B**) pre-admission fever duration, (**C**) clinical diagnosis, and (**D**) influenza virus subtype.

**Figure 3 medicina-57-01385-f003:**
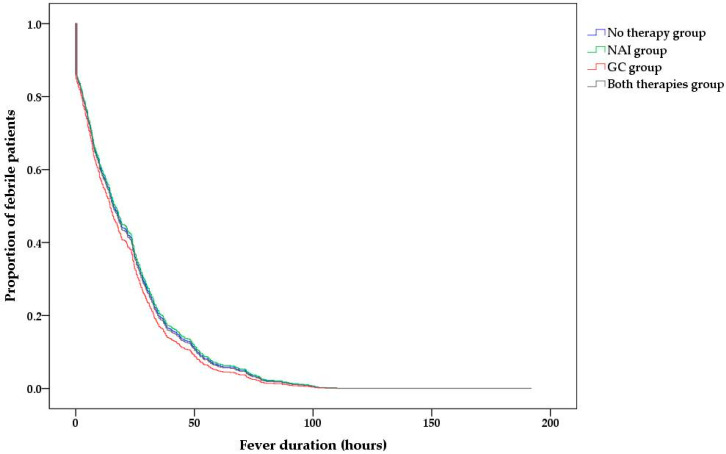
Post-admission fever duration according to the type of administered treatment.

**Table 1 medicina-57-01385-t001:** Comparison of clinical characteristics among the four treatment groups.

Factor	No Therapy Group (*n* = 52)	NAI Group (*n* = 154)	GC Group (*n* = 123)	Both Therapies Group (*n* = 272)	*p*-Value
Male sex	28 (53.8)	77 (50.0)	70 (56.9)	149 (54.8)	0.688
Age, years, median (IQR)	4 (1–7)	5 (2–9)	3 (1–6)	4 (2–7)	0.008
Influenza virus subtype					0.455
Influenza A virus	33 (63.5)	90 (58.4)	83 (67.5)	175 (64.3)	
Influenza B virus	19 (36.5)	64 (41.6)	40 (32.5)	97 (35.7)	
Diagnosis on admission					<0.001
URI	50 (96.2)	149 (96.8)	91 (74.0)	228 (83.8)	
LRI	2 (3.8)	5 (3.2)	32 (26.0)	44 (16.2)	
Final diagnosis					<0.001
URI	50 (96.2)	145 (94.2)	90 (73.2)	219 (80.5)	
LRI	2 (3.8)	9 (5.8)	33 (26.8)	53 (19.5)	
Symptoms					
Cough	47 (90.4)	136 (88.3)	107 (87.0)	247 (90.8)	0.668
Rhinorrhea	42 (80.8)	127 (82.5)	98 (79.7)	223 (82.0)	0.934
Sputum	31 (59.6)	95 (61.7)	87 (70.7)	189 (69.5)	0.189
Sore throat	3 (5.8)	24 (15.6)	14 (11.4)	24 (8.8)	0.104
Dyspnea	0 (0.0)	0 (0.0)	1 (0.8)	4 (1.5)	0.380
Vomiting	14 (26.9)	20 (13.0)	17 (13.8)	35 (12.9)	0.059
Abdominal pain	6 (11.5)	22 (14.3)	9 (7.3)	23 (8.5)	0.172
Diarrhea	6 (11.5)	18 (11.7)	11 (8.9)	19 (7.0)	0.373
Headache	8 (15.4)	24 (15.6)	8 (6.5)	18 (6.6)	0.006
Myalgia	3 (5.8)	15 (9.7)	6 (4.9)	17 (6.3)	0.388
Rash	4 (7.7)	2 (1.3)	5 (4.1)	2 (0.7)	0.005
Focal complications					
Acute otitis media	7 (13.5)	7 (4.5)	4 (3.3)	8 (2.9)	0.007
Sinusitis	1 (1.9)	2 (1.3)	1 (0.8)	2 (0.7)	0.844
Hospital days, median (IQR)	4 (4–6)	4 (4–5)	4 (4–6)	4 (4–5)	0.974
Fever duration, median (IQR)					
Before admission, days	1 (1–3)	1 (0–2)	1 (1–2)	1 (1–2)	0.101
After admission, hours	13.8 (0.1–42.0)	21.3 (7.2–33.9)	11.0 (1.2–30.5)	15.6 (4.6–31.5)	0.066
Overall, days	3 (2–5)	3 (2–4)	3 (2–4)	3 (2–4)	0.405

NAI, neuraminidase inhibitor; GC, glucocorticoid; IQR, interquartile range; URI, upper respiratory tract infection; LRI, lower respiratory tract infection.

**Table 2 medicina-57-01385-t002:** Results of the multivariate Cox regression analysis.

Factor	Hazard Ratio	95% Confidence Interval	*p*-Value
Female (vs. male)	1.072	0.909–1.263	0.410
Age, years	0.954	0.934–0.975	<0.001
Pre-admission fever, days	0.878	0.833–0.925	<0.001
Influenza A (vs. influenza B)	0.782	0.656–0.932	0.006
LRI on admission (vs. URI)	0.626	0.489–0.801	<0.001
Type of administered treatment			
No therapy	reference		
NAI therapy	1.022	0.744–1.404	0.892
GC therapy	0.951	0.682–1.328	0.769
Both therapies	0.990	0.732–1.340	0.949

LRI, lower respiratory tract infection; URI, upper respiratory tract infection; NAI, neuraminidase inhibitor; GC, glucocorticoid.

## Data Availability

The data is available upon a reasonable request to the corresponding author.

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
