# Peer review of "Effects of Antiviral Therapy and Glucocorticoid Therapy on Fever Duration in Pediatric Patients with Influenza"

_medicina, 2021, doi:10.3390/medicina57121385_

Round 1
Reviewer 1 Report
This article showed that NAI therapy and GC therapy yielded no significant effect on the fever duration in the pediatric patients with non-severe influenza. This is an important conclusion in the medical economy. I thought that this article can be an index for clinical judgment when treating patients with non-severe influenza in actual clinical practice. By the way, please tell me about the position of NAI therapy and GC therapy in the prevention of serious influenza encephalopathy.
Author Response
Thank you for your valuable comments.
Please see the attachment.

Reviewer 2 Report
This manuscript describe the comparison between the treatment outcome in paediatric patients who received NAI, GC, or both, as compared to those who received none.
I have some suggestions to further improve the manuscript:
- A table may be included to summarise the inclusion and exclusion criteria for the study, to make it clearer to the readers. The same suggestion for the patients group, it would be great to have a table to summarise the information in line 100 - 112.
- In the Definition section
- "oral oseltamivir and intravenous 118 peramivir were administered.." Does it means that the patients received both? if not, perhaps the word "and" could be replace with "or".
- Fever is generally defined as temperature more than 37.5C. Can the author explain why a temperature of 38C was chosen instead?
- Can the author explain the method of fever measurement? (infrared thermometer vs armpit vs under tongue measurement etc)
- The numbers may need to be standardised, e.g at certain places, "21" and in other places, "twenty-one"
- Line 179 "higher incidence of 179 influenza A virus infection than of influenza B virus infection" but in Table 2, it was written as "Influenza B (vs. influenza A)". This is quite confusing, and if possible to make it clearer.
- It may be useful to describe what is meant by "conservative care".
- The authors reported that "The post-admission fever duration tended to shorten with increasing age, longer pre-admission fever duration, and incidence of influenza A virus infection and lower respiratory tract infection". However, the authors also mentioned in Line 266 on the probability of empirical antibiotics' effect on the post admission fever duration in patients with LRI. If possible for the authors to clarify on this matter.
- In Line 281 - 286, the authors wrote that there are some important factors that are not following a standard guideline, such as the dose of GC. Hence, the Abstract and Conclusion section may need to be adjusted, as in my opinion, there are still areas with unclear data. Concluding that the use of GC is not necessary in non severe influenza may lead to misunderstanding on the ground, which will affect the overall clinical practice.
Thank you.
Author Response

(The authors gave the same response as above.)
